# Influence of Maternal and Paternal History of Mental Health in Clinical, Social Cognition and Metacognitive Variables in People with First-Episode Psychosis

**DOI:** 10.3390/jpm12101732

**Published:** 2022-10-18

**Authors:** Sara Mendoza-García, Helena García-Mieres, Raquel Lopez-Carrilero, Julia Sevilla-Lewellyn-Jones, Irene Birulés, Ana Barajas, Ester Lorente-Rovira, Alfonso Gutiérrez-Zotes, Eva Grasa, Esther Pousa, Trini Pelaéz, Maria Luisa Barrigón, Fermin González-Higueras, Isabel Ruiz-Delgado, Jordi Cid, Roger Montserrat, Laia Martin-Iñigo, Berta Moreno-Kustner, Regina Vila-Bbadía, Luciana Díaz-Cutraro, Marina Verdaguer-Rodríguez, Marta Ferrer-Quintero, Paola Punsoda-Puche, Paula Barrau-Sastre, Steffen Moritz, Susana Ochoa

**Affiliations:** 1Parc Sanitari Sant Joan de Déu, Sant Boi de Llobregat, 08830 Barcelona, Spain; 2Facultad de Psicología, Universidad de Barcelona, 08035 Barcelona, Spain; 3Consorcio de Investigación Biomédica en Red de Salud Mental (CIBERSAM), Instituto de Salud Carlos III, 08028 Barcelona, Spain; 4Grup MERITT, Fundació Sant Joan de Déu, Institut Sant Joan de Déu, Esplugues de Llobregat, 08950 Barcelona, Spain; 5Instituto Hospital del Mar de Investigaciones Médicas (IMIM), 08003 Barcelona, Spain; 6Consorcio de Investigación Biomedica en Red: Epidemiología y Salud Pública (CIBERESP), Instituto de Salud Carlos III, 28029 Madrid, Spain; 7Instituto de Psiquiatría y Salud Mental, Instituto de Investigación Biomédica (IdISSC), Hospital Clínico San Carlos, 28040 Madrid, Spain; 8Facultad de Psicología, Universidad Complutense de Madrid, 28223 Madrid, Spain; 9Department of Clinical and Health Psychology, Universitat Autònoma de Barcelona Cerdanyola del Vallès, 08193 Barcelona, Spain; 10Serra Húnter Programme, Government of Catalonia, 08028 Catalonia, Spain; 11Department of Research, Centre d’Higiene Mental Les Corts, 08029 Barcelona, Spain; 12Psychiatry Service, Hospital Clínico Universitario de Valencia, 46010 Valencia, Spain; 13IISPV, Hospital Universitari Psiquiàtric Institut Pere Mata, Universitat Rovira i Virgili, 43206 Reus, Spain; 14Department of Psychiatry, Institut d’Investigació Biomèdica-Sant Pau (IIB-Sant Pau), Hospital de la Santa Creu i Sant Pau, Universitat Autònoma de Barcelona, 08041 Barcelona, Spain; 15Salut Mental Parc Taulí. Sabadell (Barcelona), Hospital Universitari—UAB Universitat Autònoma de Barcelona, 08208 Barcelona, Spain; 16Neuropsiquiatria i Addicions, Hospital del Mar, IMIM (Hospital del Mar Medical Research Institute), 08003 Barcelona, Spain; 17Department of Psychiatry, IIS-Fundación Jiménez Díaz Hospital, 28040 Madrid, Spain; 18Psychiatry Service, Area de Gestión Sanitaria Sur Granada, Motril, 18600 Granada, Spain; 19Comunidad Terapéutica Jaén Servicio Andaluz de Salud, 23001 Jaen, Spain; 20Unidad de Salud Mental Comunitaria Malaga Norte, UGC Salud Mental Carlos Haya, Servicio Andaluz de Salud, 29014 Malaga, Spain; 21Mental Health & Addiction Research Group, IdiBGi Institut d’Assistencia Sanitària, 17119 Girona, Spain; 22Departamento de Personalidad, Evaluación y Tratamiento Psicológico de la Facultad de Psicología, Instituto Biosanitario de Málaga, 29010 Malaga, Spain; 23Psychology Department, FPCEE Blanquerna, Universitat Ramon Llull, 08022 Barcelona, Spain; 24Department of Psychiatry and Psychotherapy, University Medical Center Hamburg, 20251 Hamburg, Germany

**Keywords:** psychotic spectrum disorder, first degree relatives, family loading, delusion, attribution

## Abstract

This study investigates, for the first time, clinical, cognitive, social cognitive and metacognitive differences in people diagnosed with first-episode of psychosis (FEP) with and without a family history of mental disorder split by maternal and paternal antecedents. A total of 186 individuals with FEP between 18 and 45 years old were recruited in community mental-health services. A transversal, descriptive, observational design was chosen for this study. Results suggest that there is a higher prevalence of maternal history of psychosis rather than paternal, and furthermore, these individuals exhibit a specific clinical, social and metacognitive profile. Individuals with a maternal history of mental disorder scored higher in delusional experiences, inhibition of the response to a stimulus and higher emotional irresponsibility while presenting a poorer overall functioning as compared to individuals without maternal history. Individuals with paternal history of mental disorder score higher in externalizing attributional bias, irrational beliefs of need for external validation and high expectations. This study elucidates different profiles of persons with FEP and the influence of the maternal and paternal family history on clinical, cognitive, social and metacognitive variables, which should be taken into account when offering individualized early treatment.

## 1. Introduction

### 1.1. Familial Risk in Psychosis

Schizophrenia is a high-burden disorder that concurs with delusions, hallucinations, disorganized speech, disorganized/catatonic behavior or negative symptoms, causing impairment in relevant areas of functioning such as work, personal relationships or care and present for more than 6 months. Schizophrenia is one of the most prevalent psychotic diseases. It is known to have a hereditary component but also to be highly influenced by environmental triggers [1]. The estimated risk of developing schizophrenia is approximately 10% in individuals who have a mother or father with psychosis, increasing to 50% if both parents are affected, in comparison to a 1% risk in the general population [2,3]. In a study by Paruk et al. [4], 19.1% of individuals had a familial history of psychosis, while 69.9% of the sample had a first-degree family member diagnosed with a mental disorder. However, other studies found that the risk of developing schizophrenia in a first-degree relative is between 6 and 13%, decreasing to 2–4% in second-degree relatives [5].

Having a first-degree family member with psychosis is one of the individual risk predictors for developing mental or neurocognitive disorders [6,7]. This association may be due to genetic aspects but also to environmental factors such as problems in the family dynamics (relationships, affectionless control bonding style, problems in thought and behavior, conjugal difficulties and multiple hospitalizations of the family members) [8]. Familial functioning is often disrupted when one of the progenitors is affected by a severe mental disorder; however, a child that does not live with an affected progenitor can also develop psychosis [7]. 

The literature in fostered-away individuals, albeit now outdated, pointed out interesting findings in this line. As an example, Lowing et al. (1983) [9] found a significant increase in the prevalence of schizophrenia and other psychotic disorders in fostered-away individuals that had been raised without their biological parents’ influence. Biological parents pass their genetic vulnerability to psychosis on to their offspring during the first years of life, while fostered-away parents provide for a familial environment [10]. However, a study by Rosenthal et al. (2002) [11] found that fostered-away individuals with a vulnerability to schizophrenia developed psychosis under stress, although they were raised by unaffected parents. This finding suggests that a healthy and protective environment could buffer the genetic risk to schizophrenia, while an unhealthy and disruptive environment could increase it [10].

A recent meta-analysis has shown that patients who have a maternal history of psychiatric disorder have a higher risk of developing psychosis than those who have a paternal history (OR 4.60 vs. 2.73). Additionally, the risk of maternal history of psychosis increases the risk for psychosis of her son or daughter (OR = 7.61) [12]. In the same line, there is one study that focused on maternal history of mental disorders [8], observing that in a sample of high-risk children of mothers who had schizophrenia, girls presented more negative attitudes in life, such as low emotional stability and less intellectual ability. However, boys displayed more maladaptive behavior at school than males of mothers who had been diagnosed with depression or no mental disorder. 

### 1.2. Familial History and Associations to Clinical and Metacognitive Variables

The presence of symptoms directly affects community functioning in people with first-episode psychosis (FEP) [13], hence the importance of assessing its relationship with familial history. The literature examining the relationship between these variables is scarce. Both Malaspina et al. (2000) [14] and Martín-Reyes et al. (2004) [15] found that individuals with a familial history of schizophrenia have a higher prevalence of negative symptoms as well as more resistance to treatment. Recently, Canga et al. (2016) [16] described a higher prevalence of positive symptoms in those patients with a familial history of schizophrenia. Barajas et al. (2019) [17] found that familial history is one of the best predictors of the severity of positive and disorganized symptoms in people with FEP.

A growing interest in social cognition and metacognition has blossomed over the last few years [18]. It is well-known that subjects with psychosis present impairments in both constructs even in the early phases of the disease, but there is no study examining the influence of a familial history of psychosis on these constructs.

In summary, the literature exploring the role of family history of psychosis on clinical, social cognition and metacognition variables in FEP is scarce, and there is a dearth of studies examining the influence of the maternal and paternal history of psychosis independently. According to previous research, maternal and paternal history of mental disorder could differently influence the course of the disorder. The main aim of the present study was to identify differences in clinical, social cognitive and metacognitive factors between patients with maternal or paternal history of mental disorder in people with FEP.

## 2. Materials and Methods

### 2.1. Design

We performed a cross-sectional, descriptive and observational study. The design of the study stems from two research sources aimed to address the effectiveness of metacognitive training in people with FEP, under the protocol numbers NCT04429412 and NCT02340559 in https://clinicaltrials.gov/ (accessed on 5 October 2022). For this study, we only used the baseline measures of each trial.

### 2.2. Participants

The baseline measures of the trial comprised 186 outpatients with FEP between 18 and 45 years of age. Patients were recruited in one of the community mental-health services provided by the participating groups: Fundación Jiménez Díaz (Madrid), Servicio Andaluz de Salud de Jaén, Servicio Andaluz de Salud de Málaga, Centro de Salud Mental de Corporació Sanitària i Universitària Parc Taulí (Sabadell), Consultas externas del Hospital de Sant Pau (Barcelona), Centro de Higiene Mental Les Corts (Barcelona), Institut Pere Mata (Reus), Institut d’Assistència Sanitària Girona, Hospital Clínic de València and Parc Sanitari Sant Joan de Déu (PSSJD). 

Inclusion criteria were: (1) a diagnosis of schizophrenia, psychotic disorder not otherwise specified, delusional disorder, schizoaffective disorder, brief psychotic disorder, or schizophreniform disorder (according to DSM-IV-TR); (2) <5 years from the onset of symptoms; (3) a score ≥3 in delusions, grandiosity, or suspiciousness items of the Positive and Negative Syndrome Scale (PANSS) in the last year; (4) patients living in the community with clinical stability in the previous 3 months; and (5) age between 18 and 45. 

Exclusion criteria included (1) traumatic brain injury, dementia, or intellectual disability (premorbid IQ ≤ 70); and (2) substance dependence. 

### 2.3. Measures

#### 2.3.1. Sociodemographic Variables

The sociodemographic variables included were:-Familial history and other sociodemographic variables were assessed by a questionnaire designed ad hoc for the purpose of this study. The family history questionnaire assessed the presence of maternal/paternal history of mental disorder and the presence of history of psychosis in both parents. The patients were asked in the interview using this questionnaire. Their answer was checked with the electronic health records of each patient, and the referring clinician was asked for further confirmation if the information provided by the patient and the information of the electronic health records was discordant.

#### 2.3.2. Clinical Variables

The following scales were included to assess clinical state:-Structured clinical interview for DSM-IV (SCID-I) [19]: The module assessing psychosis was administered to confirm the diagnosis and double checked consulting the referral clinician and the electronic health record of the patient.-Positive and Negative Syndrome Scale (PANSS) [20,21] was used to measure positive, negative and general symptoms in psychotic disorders.-Peters Delusions Inventory (PDI) [22] was used to measure delusional experiences. The scale assesses a total of 21 items for delusional experiences and distress, preoccupation and conviction of these experiences.

#### 2.3.3. Metacognition and Social Cognition

For the assessment of metacognition and social cognition we have include the following instruments:-Beck Cognitive Insight Scale (BCIS) [23,24] was used to measure person’s insights on their judgment. The scale yields two dimensions: Self-reflectiveness and Self-certainty.-The Scale of Irrational Beliefs (TCI) [25] was used to measure irrational beliefs. The scale is composed of ten subscales: the need for external validation, high expectations, guilt, intolerance to frustration, worry and anxiety, emotional irresponsibility, problem avoidance, impotence, and perfectionism.-Internal, Personal and Situational Attributions Questionnaire (IPSAQ) [26] was used to assess attributional style in 32 different situations, attending to whether a certain situation is caused by oneself, other persons or the situation. The scale yields two subscales: personalizing bias and externalizing bias.-The Hinting Task [27,28] was used to measure theory of mind. A reduced version was used in this study. Our reduced scale is based on the items that reached better internal consistency in the Spanish validation, as the reliability of the whole scale did not reach satisfactory values. We used two research sources in this work: a subset of the sample was assessed with three stories at test and different stories at re-test to prevent learning effects. The other subset was assessed with six stories. To calculate a composite measure of the Hinting Task, we divided the total in each condition by the number of items of the test, yielding a measure between 0 and 2.-Faces Test [29,30] was used to measure emotional recognition of basic and complex emotions.

#### 2.3.4. Functional

Global Assessment of Functioning (GAF) [31] was used to measure global functioning. The scale ranges 0–100, with higher scores indicating higher global functioning.

#### 2.3.5. Neuropsychology

All the neuropsychological tests were calculated with the standardized scores. 

-The Wisconsin Sorting Card Test (WSCT) [32,33] was used to assess executive function.-Stroop Test [34] was used to measure executive function: flexibility and ability to suppress automatic responses, such as selective attention and processing speed.-The Trail Making Test (TMT-A and TMT-B) [35] was used as a measure of visuomotor attention, sustained attention, speed and cognitive flexibility.-The Continuous Performance Test (CPT-II for Windows) [36] was used to assess attention and impulsivity. Commissions and omissions scores were collected.-The Weschler Adults Intelligence Scale (WAIS) [37] subtests Vocabulary and Digits were used to measure premorbid intelligence and verbal fluency, and working memory, respectively [38].

### 2.4. Procedure

The clinicians of the participating mental health centers referred the participants that met the inclusion criteria and verbally agreed to participate in the study. The participants signed an informed consent after receiving more exhaustive information about the study and after confirmation of inclusion criteria. All instruments were administered in face-to-face interviews in the patient’s habitual mental health centers. The interviewers were general health psychologists that were trained or have long experience in administering the scale of the study and scored >0.70 in inter-rater reliability.

### 2.5. Ethics

All individuals were given an informative sheet, and all of them signed an informed consent file for participation in this study. The Ethics Committee of each participating center approved this project, and the investigation was carried out in accordance with the Declaration of Helsinki, as revised in 2013.

### 2.6. Statistical Analyses

The statistical analysis was conducted using SPSS (version 22). The data is presented with descriptive statistics. 

For the main aim of the study, we used Student’s *t*-tests and Mann–Whitney U-test (non-parametric data) to compare patients with presence and no presence of a history of mental disorder (for father and mother). The effect size was calculated with Cohen’s d. Additionally, we also created a variable including four categories: no presence of history of mental disorder in father or mother, history of mental disorder in mother, history of mental disorder in father or history of mental disorder in both. We did not use correction for multiple comparisons given the exploratory nature of this study [39].

## 3. Results

### 3.1. Sociodemographic Variables

Table 1 displays the sociodemographic characteristics of the sample. There were twice as many males in the sample than females. Most of the sample was single (80%), with secondary education completed, and actively working (41%). Considering diagnosis, 40% of the sample was diagnosed with schizophrenia and 30% was diagnosed with psychotic disorder not otherwise specified. A total of 23% had a maternal history of mental disorder (*n* = 42), 10% had a paternal history of mental disorder (*n* = 18) and 2% had both parental and maternal history of mental health. Of them, 18 participants have maternal or paternal history of mental disorder of psychosis, 15 of them of have depression, dysthymia or bipolar disorder, 3 have anxiety, 2 have personality disorders and 11 have another diagnosis. Among the individuals with a maternal history of mental disorder, 13 of them (31%) had a maternal history of psychosis, while five participants (28%) had a paternal history of psychosis. No significant differences were found in the prevalence of a maternal and paternal history of mental disorders between women and men.

### 3.2. Clinical Variables

Table 2 shows differences in psychotic symptoms between the individuals presenting a maternal and paternal family history of mental disorder. Participants referring maternal psychiatric history scored significantly higher in the PDI subscales: presence of delusional experiences, distress, preoccupation and conviction. The effect size in all the variables was medium (d > 0.5), suggesting the practical significance of these results. Furthermore, those individuals referring to maternal psychiatric history scored lower in general functioning as measured by the GAF. There was no association between having a paternal history of mental disorder and symptoms. 

When we compared four groups, we found that those patients with maternal history of mental disorder score lower in ranks regarding GAF assessment (*p* = 0.032). The mean of GAF in maternal history was 55.35 (SD = 14.08), while no presence of mother or father with history was 60.55 (11.39), father 58.5 (SD = 14.73) and both 56.67 (SD = 56.67). Moreover, we found that those with maternal history of mental disorder score higher ranks in the conviction subscale of the PDI (*p* = 0.019) and a tendency in PDI total (*p* = 0.075), anxiety subscale (*p* = 0.062) and preoccupation subscale of PDI (*p* = 0.064).

### 3.3. Cognitive Variables

Table 3 shows differences in cognitive performance between participants with a maternal and paternal history of mental disorder. Individuals presenting a maternal history of mental disorder scored significantly higher in CPT commissions (*p* = 0.036, d = 0.49). There were no significant differences in TMT-A, TMT-B or estimated IQ. Furthermore, we did not find any association between having a paternal history of mental disorder and cognitive functioning. 

### 3.4. Metacognitive Variables

Table 4 summarizes the differences in metacognition and social cognition. Individuals with maternal psychiatric history scored higher in TCI emotional irresponsibility, with a 0.49 medium effect size. Regarding IPSAQ scores, individuals with paternal psychiatric history had higher scores in externalizing bias, less need for external validation and lower expectations. We did not find any significant difference in maternal or paternal psychiatric history in theory of mind (Hinting Task Scores), emotion recognition (Faces Test) and cognitive insight (BCIS).

Comparing four groups, we have found that those patients with paternal history of mental disorder have a tendency to score higher in ranks regarding externalizing bias (*p* = 0.080). On the other hand, we have found that those patients with maternal history of mental disorder have a higher score in the subscale of higher expectations than the other groups (*p* = 0.022). Additionally, this group has also a tendency to score high in helplessness (*p* = 0.071).

## 4. Discussion

Consistent with our hypothesis, we found significant differences in clinical, cognitive, social cognitive and metacognitive variables between patients with a maternal or paternal history of mental disorder and those without one. 

We found a higher prevalence of a maternal history of mental disorder coinciding with previous literature [12]. A study by Goldstein et al. (2013) [40] reported that heritability of psychosis is linked to the X chromosome, which could be one explanation for our findings. Another explanation is that women with psychosis tend to have a later age of onset than men, which permits them to find a stable partner and form a family before symptoms appear. In comparison, men often have an earlier onset of the disease, which causes a direct impact on their social environment and partnership [41,42,43,44]. The percentage of familial history in our sample was similar to the one reported by Canga et al. (2016) [16] (33% vs. 18.5%). In addition, consistent with a review from our group [43], we did not find females to have a higher prevalence of a familial history of mental illness than males. 

Barajas et al. (2019) [17] found that a family history of mental illness is one of the best predictors of positive and disorganized symptoms. Similarly, our study only yielded significant differences in the whole scale and subscales of PDI in those with a maternal history of mental disorder, even comparing with the presence of both maternal and paternal history. In contrast to our results, Paruk et al. (2017) [4] found that individuals with familial history showed significantly less positive symptoms than those without. However, this sample was not split by maternal and paternal psychiatric history, but according to our results, future studies should examine family history of mental illness separately, as it seems that maternal but not paternal mental illness is what may influence symptoms. 

Regarding neuropsychological performance, we found no major differences between people with paternal or maternal symptoms. We found that participants with a maternal history of mental illness were less accurate in inhibiting responses to stimuli (commissions), which is consistent with Tang et al. (2018) [45], who found that more insight in mothers was negatively associated to impulsivity in their offspring. However, this study was not focused on psychosis but in the general population, and it did not control for mental disorders, which hampers the comparability of results. 

We found a higher prevalence of externalizing biases but less presence of personalizing biases in patients with a paternal history of psychosis, even after comparison with other groups. 

These results were not consistent with other studies [46,47], which reported that persons with psychosis tend to attribute negative events to external causes, especially to other people. This concept is relevant given the relationship of these biases with the formation and maintenance of delusions [48]. However, these studies did not analyze maternal and paternal history separately, so future studies could yield more valuable results if they succeed at highlighting different attributional patterns as a function of familial history, especially in those having paternal psychiatric history and its relationship with delusions. 

When examining irrational beliefs, we found that individuals with a paternal history of mental illness need less validation from others but at the same time had lower expectations. Individuals with a maternal history of mental disorder had more emotional irresponsibility. Taken together, our findings seemed to suggest that maternal and paternal psychiatric history affects personal beliefs differently. The Attachment Theory postulates that there is a need to develop a solid bond with the mother, which will influence future interpersonal relations [49]. Having a mother with a mental disorder may decrease the chances of building a safe bond, and the lack of thereof may in turn influence stress regulation and emotional processes [50] in the early childhood that could cause these problems to be more evident in the adolescent period. Parker et al. (1982) [51] offered a different point of view and proposed that individuals with schizophrenia consider their fathers as more overprotective. Having a mental disorder may foster an overprotective upbringing, which can generate less interest in the person’s environment and one’s own beliefs toward one’s possibilities. High expressed emotion has previously been related to an earlier onset of schizophrenia [52].

### 4.1. Limitations

First, even if our sample had an adequate size, the proportion of individuals with a familial history of the disorder was low. Thus, our study could lack sufficient statistical power and be vulnerable to type II errors, especially for the comparison of participants with and without paternal history of mental disorder [5]. However, this is an exploratory analysis, and further research with higher samples should be performed. Second, since parental history of mental disorders was assessed with an ad hoc questionnaire, it is possible that some patients might not had the correct information, and while that bias was controlled by checking electronic health records and clinicians, family members were not directly asked. Third, the cross-sectional design of our study impedes to establish causal associations. We also only had one general measure of functioning (GAF), and other measures of neuropsychology assessment could be also included. Despite being widely used in research, it fails to cover all nuances of functional outcome. Finally, we did not have a healthy control group. Future studies should include follow-ups, more nuanced measures of social functioning, and a control group without mental disorders to further validate or reject our findings. 

### 4.2. Therapeutic Implications

Even though our study suggests different clinical and therapeutic implications, our main interest is to raise awareness on an early and specific treatment on the group of high-risk individuals whose mothers and fathers have a history of mental illness. An interesting approach would be to design personalized treatments that target symptoms, social cognition, and metacognition depending on the affected relative. The algorithms for the detection of psychosis should include different weights for people with fathers or mothers presenting a history of mental disorder. In this line, the existing program KIDSTIME [53] consists of monthly sessions for mothers, fathers, and children aged 5–15 with parents affected with a mental disorder. The objective of this intervention is to facilitate a group setting in which psychoeducational techniques, drama, and leisure activities are combined to improve the familial bond and decrease the stigmatizing attitudes toward mental disorder. Similarly, mother and baby units could be useful to alleviate the problems associated with being a women with schizophrenia, such as single parenting, substance abuse, domestic violence and poverty [54,55], and to increase attachment [56].

## 5. Conclusions

Familial history of psychosis among first-degree relatives is considered a prominent individual risk predictor for developing a mental disorder, and therefore, it should be better studied. We have found evidence for different clinical, cognitive, social cognitive and metacognitive affectations as a function of having a mother or a father with a history of mental illness. Our findings highlight the importance of assessing an affected person’s progenitors, as they could offer key information about the dynamics that may be playing a role in adolescents with mental health problems and to tailor an adequate treatment to their carers [4]. Assessing social cognition and metacognition on top of psychotic symptomatology in early psychosis can be crucial to identify individuals that could benefit from intensive therapeutic approaches [38].

More awareness of this field is urged, as further studies will allow us to compare results and present more specific differences in symptoms, metacognition and social cognition in individuals with a history of mental disorder in their first-degree family members. Further knowledge will, in turn, allow us to design more specific detection methods and interventions considering these differences.

## Figures and Tables

**Table 1 jpm-12-01732-t001:** Sociodemographic characteristics of the sample.

Variables	Categories	*n*	%
Gender	Men	125	67
	Women	60	32
Marital status	Single	157	84
	Married	15	8
	Separated	12	6
	Widow	2	1
Educational level	Primary	48	26
	Secondary	91	49
	University	47	25
Employment	Work	77	41
	Student	31	17
	Long sick leave	41	22
	Unemployed	34	18
	Others	3	2
Diagnosis	Schizophrenia	74	40
	Brief psychotic disorder	16	9
	Non-specified psychotic disorder	54	30
	Schizoaffective disorder	18	10
	Delusional disorder	11	6
	Schizophreniform disorder	10	6
		*M*	*SD*
Age		27.95	7.38
Number of psychiatric admissions		1.22	1.41

**Table 2 jpm-12-01732-t002:** Differences between maternal and paternal history of mental disorder and psychotic symptoms and functioning.

	Maternal History of Mental Disorder				Paternal History of Mental Disorder			
	Presence	Absence			Presence	Absence		
	*M (SD)*	*M (SD)*	*p Value **	*d ****	*M (SD)*	*M (SD)*	*p Value ***	*d ****
Positive PANSS	14.5 (5.81)	13.86 (5.88)	0.535	0.11	14.94 (5.02)	13.91 (5.93)	0.491	0.19
Negative PANSS	15.24 (6.03)	15.01 (6.43)	0.836	0.04	14.00 (5.14)	15.17 (6.43)	0.470	0.20
General PANSS	30.60 (7.65)	29.01 (8.91)	0.297	0.19	29.41 (5.02)	29.37 (8.94)	0.983	0.01
Total PDI	8.25 (5.51)	5.38 (4.21)	**0.004**	0.59	6.76 (5.15)	5.92 (4.61)	0.494	0.17
Distress PDI	23.07 (24.11)	12.39 (11.78)	**0.031**	0.56	18.65 (21.23)	14.26 (15.11)	0.300	0.24
Preoccupation PDI	22.71 (24.21)	12.78 (13.01)	**0.046**	0.51	21.00 (21.24)	14.12 (15.83)	0.117	0.37
Conviction PDI	26.75 (24.61)	15.45 (14.15)	**0.027**	0.56	24.59 (22.14)	17.01 (16.75)	0.102	0.39
GAF	55.67 (13.53)	60.33 (11.60)	**0.029**	0.37	57.35 (12.26)	59.45 (12.20)	0.501	0.17

PANSS: Positive and Negative Syndrome Scale; PDI: Peters Delusion Scale; GAF: Global Assessment Scale. * Student’s *t*-test, ** Mann–Whitney U-test, *** Cohen’s d.

**Table 3 jpm-12-01732-t003:** Relationship between maternal and paternal history of mental disorder and cognition.

		Maternal History of Mental Disorder				Paternal History of Mental Disorder			
		Presence	Absence			Presence	Absence		
		*M (SD)*	*M (SD)*	*p Value*	*d*	*M (SD)*	*M (SD)*	*p Value*	*d*
CPT	Omissions	71.29 (41.09)	96.05 (111.63)	0.272	0.29	71.34 (44.78)	93.09 (105.72)	0.436	0.26
	Commissions	58.89 (12.00)	52.93 (12.57)	0.036	0.49	59.37 (14.70)	53.56 (12.17)	0.099	0.43
STROOP	Word	43.49 (9.35)	40.15 (10.28)	0.070	0.34	43.19 (9.96)	40.66 (10.17)	0.343	0.25
	Color	38.28 (10.31)	37.04 (9.80)	0.490	0.12	36.69 (8.22)	37.38 (10.07)	0.792	0.08
	Interference	53.31 (10.85)	54.25 (11.09)	0.639	0.09	52.75 (9.39)	54.17 (11.19)	0.624	0.14
WCST	Correct hits	73.81 (12.91)	72.82 (11.85)	0.659	0.08	69.88 (13.04)	73.39 (11.94)	0.256	0.28
	Total errors	42.95 (7.81)	44.38 (10.61)	0.446	0.15	40.35 (8.23)	44.48 (10.16)	0.109	0.45
	Perseverative errors	44.14 (7.51)	45.44 (11.03)	0.500	0.14	40.76 (8.47)	45.65 (10.45)	0.065	0.51
	Non-perseverative errors	42.05 (8.25)	44.14 (10.40)	0.264	0.22	42.00 (6.27)	43.87 (10.31)	0.467	0.22

CPT: Continuous Performance Test; WCST: Wisconsin Card-Sorting Test.

**Table 4 jpm-12-01732-t004:** Relationship between maternal and paternal history of mental disorder and metacognition and social cognition.

		Maternal History of Mental Disorder				Paternal History of Mental Disorder			
		Presence	Absence	*p Value **	*d ****	Presence	Absence	*p Value ***	*d ****
		*M (SD)*	*M (SD)*			*M (SD)*	*M (SD)*		
Attribution IPSAQ scale	Personalizing bias	1.33 (0.86)	1.20 (0.58)	0.282	0.18	1.12 (0.50)	1.24 (0.67)	0.487	0.20
	Externalizing bias	0.48 (3.94)	1.03 (3.69)	0.397	0.14	2.78 (4.05)	0.71 (3.67)	0.026	0.54
TCI	Emotional irresponsibility	23.11 (8.50)	19.46 (6.29)	0.043	0.49	21.22 (7.30)	20.15 (6.97)	0.549	0.15
	Need for external validation	24.57 (7.16)	23.98 (5.41)	0.639	0.09	21.39 (5.22)	24.59 (5.83)	0.031	0.58
	High expectations	18.25 (4.12)	16.88 (4.14)	0.127	0.33	15.00 (4.90)	17.58 (3.92)	0.014	0.58

IPSAQ: Internal, Personal and Situational Attributions Questionnaire; TCI: Scale of Irrational Beliefs. * Student’s *t*-test, ** Mann–Whitney U-test, *** Cohen’s d.

## Data Availability

Any required data/information about the manuscript will be available upon reasonable request to the corresponding author.

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
