# Peer review of "Influence of Maternal and Paternal History of Mental Health in Clinical, Social Cognition and Metacognitive Variables in People with First-Episode Psychosis"

_jpm, 2022, doi:10.3390/jpm12101732_

Round 1
Reviewer 1 Report
Thank you for the opportunity to review this paper that I find very interesting.
I have a few comments that could improve the paper:
1. There are some language issues, e.g., in the introduction: The estimated risk of developing schizophrenia is of approximately... and other places in the paper.
2. In the introduction, the authors say that ‘family member with psychosis is the best individual risk predictor...’ I would use a different word like most significant because the word has a positive connotation.
3. I miss some definitions in the introduction, e.g., a short definition of schizophrenia (according to DSM because I see you use SCID?), and a short introduction to psychosis.
4. The authors mention ‘psychiatric disorders’ several times in the paper, and present results including ‘history of mental disorder’. The internal differences between mental disorders are quite large, and some affect individuals more than others – a short depressive episode would likely have a smaller impact on relatives than a non-treated borderline diagnosis, just to give an example. I think that a more nuanced description of ‘mental disorder’/’psychiatric disorders’ would be helpful. Do they have any information on, what kind of mental disorders, the parents present with? Have the parents been diagnosed and treated? When did they suffer from these disorders?
5. On the top of page 3, I don´t understand the word ‘propand’? And the sentence that starts with ‘However, boys displayed a more aggressive...’ is also unclear to me. I think I know what you mean by ‘community functioning’ but I would use another word like social functioning or describing what you mean by community functioning.
6. The wording in the beginning of ‘procedure’ is unclearly formulated (starts with ‘after receiving more exhaustive…).
7. In Table 1, you mention the categories ‘disability’ and ‘inactive’ under employment – it would be good if it was clearer what you mean by these categories, either by renaming them, or by explaining in a table footnote.
8. I miss a description of the cognitive tests that are used in the study. They are mentioned only in Table 3.
9. In the discussion, you mention a later onset of disease as another explanation, because they might have formed a family before appearance of symptoms. I´m not sure how this explains the results, so it would be good to clarify.
10. It would be interesting to hear more thoughts on how the results could be understood. You mention attachment theory and you could elaborate more on that. How is the connection between mother and child possibly disrupted or affected by mental disorders (again, a differentiation between severity of mental disorders would be appropriate)? Do you have any thoughts on gender differences, whether girls/women are more affected by maternal mental disorders than boys/men? In depression research, some authors argue that there are gender differences (e.g., https://books.google.com/books?hl=da&lr=&id=Cfl5AgAAQBAJ&oi=fnd&pg=PP36&dq=maternal+depression+affects+girls&ots=UQK0qQo5fF&sig=LuMhxH4BOjGfjg8D3aDM_c2H44E#v=onepage&q=maternal%20depression%20affects%20girls&f=false).
11. The first sentence in the ‘limitation’ section is unclear.
Author Response
Reviewer 1:
Thank you for the opportunity to review this paper that I find very interesting.
I have a few comments that could improve the paper:
- There are some language issues, e.g., in the introduction: The estimated risk of developing schizophrenia is ofapproximately... and other places in the paper.
Thank you for your comments regarding language issues, we have review the manuscript by an expert English.
- In the introduction, the authors say that ‘family member with psychosis is the bestindividual risk predictor...’ I would use a different word like most significant because the word has a positive connotation.
We agree with the reviewer, and we have modified the word “best” for “one of the …”
- I miss some definitions in the introduction, e.g., a short definition of schizophrenia (according to DSM because I see you use SCID?), and a short introduction to psychosis.
We have include a definition of schizophrenia in the introduction section:
“Schizophrenia is a high burden disorder that include delusions, hallucinations, disorganized speech, disorganized/catatonic behavior or negative symptoms, causing impairment in relevant areas of functioning such as work, personal relationships or care and present for more than 6 months. Schizophrenia is one of the most prevalent psychotic diseases. It is known to have a hereditary component, but also to be highly influenced by environmental triggers.”
- The authors mention ‘psychiatric disorders’ several times in the paper, and present results including ‘history of mental disorder’. The internal differences between mental disorders are quite large, and some affect individuals more than others – a short depressive episode would likely have a smaller impact on relatives than a non-treated borderline diagnosis, just to give an example. I think that a more nuanced description of ‘mental disorder’/’psychiatric disorders’ would be helpful. Do they have any information on, what kind of mental disorders, the parents present with? Have the parents been diagnosed and treated? When did they suffer from these disorders?
We have included information about the percentage of mental disorder diagnosis of the parents:
“Of them, 18 participants have maternal or paternal history of mental disorder of psychosis, 15 of them of depression, dysthymia or bipolar disorder, 3 anxiety, 2 personality disorders and 11 other diagnosis. “
However, we do not have information about the confirmation of diagnosis and years of evolution.
- On the top of page 3, I don´t understand the word ‘propand’? And the sentence that starts with ‘However, boys displayed a more aggressive...’ is also unclear to me. I think I know what you mean by ‘community functioning’ but I would use another word like social functioning or describing what you mean by community functioning.
We have modified the word proband for “son or daughter”.
We have modified also the sentence “displayed a more aggressive…” for “maladaptive behavior”
- The wording in the beginning of ‘procedure’ is unclearly formulated (starts with ‘after receiving more exhaustive…).
We have modified the sentence
“The participants signing the informed consent after receiving more exhaustive information about the study and after confirmation of inclusion criteria”
- In Table 1, you mention the categories ‘disability’ and ‘inactive’ under employment – it would be good if it was clearer what you mean by these categories, either by renaming them, or by explaining in a table footnote.
|
We have change “disability” by “Long sick leave”, and “inactive” by “unemployed” |
- I miss a description of the cognitive tests that are used in the study. They are mentioned only in Table 3.
Thank you to the reviewer for this important mistake.
We have include the neuropsychology assessment in the method section:
“Neuropsychology:
All the neuropsychological tests were calculated with the standardized scores.
-The Wisconsin Sorting Card Test (WSCT) (Grant and Berg, 1948; Tien et al., 1996) was used to assess executive function.
-Stroop Test (Stroop, 1935) was used to measure executive function: flexibility and ability to suppress automatic responses, like selective attention and processing speed.
-The Trail Making Test (TMT-A and TMT-B) (Sánchez-Cubillo et al., 2009) was used as a measure of visuomotor attention, sustained attention, speed and cognitive flexibility.
-The Continuous Performance Test (CPT-II for Windows) (Spreen and Strauss, 1998) was used to assess attention and impulsivity. Commissions and omissions scores were collected.
- The Weschler Adults Intelligence Scale (WAIS) (Wechsler, 1999) subtests Vocabulary and Digits were used to measure premorbid intelligence and verbal fluency, and working memory respectively (González-Blanch et al., 2011).”
- In the discussion, you mention a later onset of disease as another explanation, because they might have formed a family before appearance of symptoms. I´m not sure how this explains the results, so it would be good to clarify.
We think that there is a high prevalence of mothers with psychosis than men because women start the illness later and they have the opportunity to have children. However, most of the men have an earlier onset of the illness, having great problems to have a partner and children. In fact, the prevalence of motherhood in women with psychosis is around 50% while only 5% of parents have children.
We have modified the sentence in order to better understanding:
“However, another explanation could be that women with psychosis had a later onset of the disease than men, which may have allowed them to have a better adaptation in community and to have partner and a family before the appearance of symptoms”
- It would be interesting to hear more thoughts on how the results could be understood. You mention attachment theory and you could elaborate more on that. How is the connection between mother and child possibly disrupted or affected by mental disorders (again, a differentiation between severity of mental disorders would be appropriate)? Do you have any thoughts on gender differences, whether girls/women are more affected by maternal mental disorders than boys/men? In depression research, some authors argue that there are gender differences (e.g., https://books.google.com/books?hl=da&lr=&id=Cfl5AgAAQBAJ&oi=fnd&pg=PP36&dq=maternal+depression+affects+girls&ots=UQK0qQo5fF&sig=LuMhxH4BOjGfjg8D3aDM_c2H44E#v=onepage&q=maternal%20depression%20affects%20girls&f=false).
We have explained better this part in the manuscript:
“Having a mother with a mental disorder may decrease the chances of building a safe bond, and the lack of thereof may in turn influence stress regulation and emotional processes in the early childhood that could cause this problems more evident in the adolescent period.”
In our case, both women and men with psychosis have the same prevalence of maternal and parental presence of disorders as we have indicated in the manuscript.
- The first sentence in the ‘limitation’ section is unclear.
We agree with the reviewer and we have deleted this sentence.
Reviewer 2 Report
The authors investigated the clinical, cognitive, social cognitive and metacognitive differences in patients with first-episode of psychosis with and without a family history of mental disorder split by maternal and paternal antecedents. Overall this is an interesting study. The paper has the potential to contribute to the existing scientific literature in this area. I have a few comments to further improve the quality of the authors’ paper. I have outlined these issues below:
1. A cross-sectional, descriptive and observational study design, using baseline measures of participants of two trials, and lacking of healthy individuals as a control group are major limitations of the study. Did the authors analyze the dose-dependent effects of parental history of mental health on clinical, cognitive, social cognitive and metacognitive variables? Four groups were necessary for the analyses. (1) Only maternal history of mental health, (2) Only paternal history of mental health, (3) Both maternal and paternal history of mental health, and (4) Neither maternal or paternal history of mental health. It could be an interesting result if there were any significant finding.
2. The study lacks sufficient statistical power and is vulnerable to type II error. This is especially possible for the comparison of participants with and without paternal history of mental health.
3. Please justify why the authors did not use correction for multiple comparisons given the exploratory nature of this study. For example, this is a pilot study. If not, at least FDR method can be used for the correction of multiple testing. This is especially important when data-driven research is carried out.
4. Did the authors analyze whether the differences in metacognition, social cognition and cognition between patients with first-episode of psychosis with and without a family history of mental disorder split by maternal and paternal antecedents were affected by the treatment they received (e.g., antipsycotic drugs, sedative-hypnotics, or rehabilitation program)
5. Did the authors analyze the other indices of CPT? For example, hit RT, Variability, and Detectability (d'). Were there differences between patients with first-episode of psychosis with and without a family history of mental disorder split by maternal and paternal antecedents?
In the reviewer’s opinion, the above-mentioned issues need to be addressed by the authors.
Author Response
The authors investigated the clinical, cognitive, social cognitive and metacognitive differences in patients with first-episode of psychosis with and without a family history of mental disorder split by maternal and paternal antecedents. Overall this is an interesting study. The paper has the potential to contribute to the existing scientific literature in this area. I have a few comments to further improve the quality of the authors’ paper. I have outlined these issues below:
- A cross-sectional, descriptive and observational study design, using baseline measures of participants of two trials, and lacking of healthy individuals as a control group are major limitations of the study. Did the authors analyze the dose-dependent effects of parental history of mental health on clinical, cognitive, social cognitive and metacognitive variables? Four groups were necessary for the analyses. (1) Only maternal history of mental health, (2) Only paternal history of mental health, (3) Both maternal and paternal history of mental health, and (4) Neither maternal or paternal history of mental health. It could be an interesting result if there were any significant finding.
We agree with the reviewer that the influence of maternal and paternal history of mental health problems should be considered as another group. Following the suggestions of the reviewer, we have performed an Anova analysis for non-parametric measures (Kruskall-Wallis) comparing four groups: no maternal and paternal history of mental health, maternal, paternal and both. We have decided to use non-parametric analyses due to the non-normal distribution of the data.
We have included several modifications in the manuscript:
Method:
“Additionally, we also created a variable including four categories: no presence of history of mental disorder in father or mother, history of mental disorder in mother, history of mental disorder in father or history of mental disorder in both.”
Results:
“A total of 23% had a maternal history of mental disorder (n=42),10% had a paternal history of mental disorder (n=18) and 2% had both parental and maternal history of mental health.”
“When we compared four groups, we found that those patients with maternal history of mental health score lower in ranks regarding GAF assessment (p=0.032). Mean of GAF in maternal history was 55.35 (SD=14.08) while no presence of mother or father with history was 60.55 (11.39), father 58.5 (SD=14.73) and both 56.67 (SD=56.67). Moreover, we found that those with maternal history of mental health score higher ranks in the conviction subscale of the PDI (p=0.019) and a tendency in PDI total (p=0.075), anxiety subscale (p=0.062) and preoccupation subscale of PDI (p=0.064).”
“Comparing four groups we have found that those patients with paternal history of mental have a tendency to score higher in ranks regarding externalizing bias (p=0.080). On the other hand, we have found that those patients with maternal history of mental health have a higher score in the subscale of higher expectations than the other groups (p=0.022). Additionally, this group has also a tendency to score high in helplessness (p=0.071)”.
Discussion section:
“In this line, our study only yielded significant differences in the whole scale and subscales of PDI in those with a maternal history of mental disorder, even comparing with presence of both maternal and paternal history.”
“A higher prevalence of externalizing bias and less presence of blaming others for failure was found in patients with a paternal history of psychosis, even comparing with the other groups.”
- The study lacks sufficient statistical power and is vulnerable to type II error. This is especially possible for the comparison of participants with and without paternal history of mental health.
We agree with the reviewer; however, this is an exploratory analysis in order to develop further research considering this limitation.
We have included a sentence in the limitation section explaining this problem:
“First, our sample have an adequate size, however the proportion of individuals with a familial history of the disorder was low. In this line, the study could lacks of sufficient statistical power and is vulnerable to type II error, especially for the comparison of participants with and without paternal history of mental health. However this is an exploratory analysis and further research with higher samples should be performed”
- Please justify why the authors did not use correction for multiple comparisons given the exploratory nature of this study. For example, this is a pilot study. If not, at least FDR method can be used for the correction of multiple testing. This is especially important when data-driven research is carried out.
As we have commented, we did not use correction for multiple comparisons given the exploratory nature of this study as suggested Bender, R., & Lange, S. (2001). This is a pilot study and further research should be done in order to confirm our results.
- Did the authors analyze whether the differences in metacognition, social cognition and cognition between patients with first-episode of psychosis with and without a family history of mental disorder split by maternal and paternal antecedents were affected by the treatment they received (e.g., antipsycotic drugs, sedative-hypnotics, or rehabilitation program)
Thank you to the reviewer for this interesting suggestion. We have included antipsychotic drugs as a covariant in the analyses, however we did not obtain different results and this variable was not significant.
- Did the authors analyze the other indices of CPT? For example, hit RT, Variability, and Detectability (d'). Were there differences between patients with first-episode of psychosis with and without a family history of mental disorder split by maternal and paternal antecedents?
The common information between the two projects for what respects the CPT was only the indices of omissions and commissions. It should be interesting to include these indexes in the future.
We have included a sentence in the limitation section
“We also only had one general measure of functioning is (GAF) and other measures of neuropsychology assessment could be also included.”
Round 2
Reviewer 2 Report
The authors have substantially improved their manuscripts. I have no further comments.